# Food Categories for Breakfast and Mental Health among Children in Japan: Results from the A-CHILD Study

**DOI:** 10.3390/nu15051091

**Published:** 2023-02-22

**Authors:** Yukako Tani, Aya Isumi, Satomi Doi, Takeo Fujiwara

**Affiliations:** 1Department of Global Health Promotion, Tokyo Medical and Dental University, 1-5-45 Yushima, Bunkyo-ku, Tokyo 113-8519, Japan; 2Department of Health Policy, Tokyo Medical and Dental University, 1-5-45 Yushima, Bunkyo-ku, Tokyo 113-8519, Japan

**Keywords:** breakfast, rice, bread, children, behavioral problems, mental health

## Abstract

There is limited evidence that what children eat for breakfast contributes to their mental health. This study aimed to examine the associations between food categories for breakfast and mental health among children in Japan. A proportion of participants aged 9–10 years in the Adachi Child Health Impact of Living Difficulty (A-CHILD) study in Japan who consumed breakfast daily were included (*n* = 281). Foods eaten for breakfast were reported by the children each morning for 7 consecutive days, and defined according to the food categories in the Japanese Food Guide Spinning Top. Child mental health was assessed by caregivers using the Strength and Difficulties Questionnaire. The mean intake frequencies per week were six times for grain dishes, two times for milk products, and one time for fruits. Linear regression analysis revealed an inverse association between the frequent intake of grain dishes, whether rice or bread, and problem behaviors after adjustment for potential confounders. However, confectionaries, which mainly consisted of sweet breads or pastries, were not associated with problem behaviors. The intake of non-sweet grain dishes at breakfast may be effective for preventing behavioral problems in children.

## 1. Introduction

Mental disorders in children are a major cause of health-related burdens, with approximately one-fifth of children suffering from mental disorders [1,2]. In a 2015 meta-analysis, the estimated worldwide pooled prevalence of individual mental disorders in children, such as any anxiety, depressive disorder, attention-deficit/hyperactivity disorder, or conduct disorder, ranged from 2% to 7% [3]. A recent systematic review evaluating the effect of COVID-19 lockdowns on children’s mental health reported a substantial increase in mental disorders after the lockdowns compared with before the lockdowns [4]. Mental health problems in children can lead to long-term negative effects during the life course, including poor economic outcomes, injury risk, and premature death [5,6,7]. Moreover, a significant proportion of mental health problems in adults begin during early life [8,9]. Therefore, it is important to identify preventive factors that can be targeted by interventions from childhood.

Breakfast is one of the potential preventive factors for children’s mental health. Breakfast is generally defined as the first meal of the day and eaten within 2 h of waking up, typically no later than 10:00 a.m. [10]. Breakfast interrupts the overnight fast and provides fuel to the brain and body [11]. Compared with adults, children may be more susceptible to the adverse effects of brief fasting because of the greater metabolic demands of the brain relative to the liver and muscle glycogen stores and gluconeogenic capacity [12]. The most investigated benefit of breakfast for children is its association with cognitive performance [13,14,15]. A systematic review showed that several tasks requiring attention, executive function, and memory were more reliably facilitated after breakfast consumption relative to fasting [13]. While there is some evidence for an association between breakfast skipping and mental disorders in children [16,17,18], the association may also depend on the quality of the breakfast [19,20]. High-quality breakfasts, such as those including bread, dairy products, and fruits but not confectionaries, were associated with better mental health, while low-quality breakfasts were associated with poorer mental health than breakfast skipping [19,20]. These findings suggest that the types of food eaten for breakfast are more important than whether or not breakfast is eaten.

Meals vary by culture and should be individually considered. Generally, breakfast has less variety and the same foods are eaten, while lunch and dinner have greater variety [21]. Western breakfasts are relatively simple, with three recommended food groups: cereals, dairy products, and fruits [22]. Thus, in a study conducted in Western culture settings, a high-quality breakfast is defined as the consumption of bread/cereals and/or dairy products and no commercially baked goods (e.g., biscuits/pastries) [20]. Another study defined a high-quality breakfast as the consumption of three or more of the following five food groups: bread/cereals, vegetables, fruits, dairy products, and meat/meat alternatives [19].

In Japan, there is no clear definition of a high-quality breakfast. Japanese breakfasts are relatively complex and have more variety than western breakfasts [23]. The Japanese diet underwent dramatic changes and became more Westernized between 1950 and 1975, characterized by the increased intake of milk, meat, poultry, eggs, and fat, and decreased intake of barley, potatoes, and rice [24,25]. Interestingly, the foods consumed in Japan have varying degrees of Westernization by meal types. For example, bread and dairy products are mainly eaten at breakfast rather than at lunch and dinner [26]. The ideal Japanese breakfast is considered to be derived from traditional Japanese cuisine (e.g., rice and miso soup) with nutritionally recommended items such as salads, but there is a gap between the ideal meal and reality [27]. Two main staple options for breakfast exist in Japan, rice-based and bread-based, with those preferring each option described as “gohan-ha” and “pan-ha”, respectively [23,26]. A study analyzing breakfast meal patterns among Japanese adults reported four types of patterns, including two staple meal patterns, as follows: rice-based (rice/vegetable/pulse/seasoning), bread-based (bread/dairy/fruit/sugar), meat/egg/fat, and tea/coffee [23]. However, the food types eaten by children for breakfast and their associations with mental health among children are not well understood.

Therefore, the aim of the present study was to investigate the types of food eaten for breakfast by children in Japan and to determine their relationships with mental health. There is consensus on the importance of eating breakfast; however, there is limited evidence that what children eat for breakfast contributes to their mental health. The goal of this study was to contribute to the improvement of children’s mental health by providing evidence about what to eat for breakfast to prevent problem behaviors in children.

## 2. Materials and Methods

### 2.1. Study Design and Subjects

This study used data on dietary habits and health from the Adachi Child Health Impact of Living Difficulty (A-CHILD) project, which was initiated in 2015 to evaluate the determinants of health among children in Adachi, Tokyo, Japan [28]. Specifically, the study used adjunct data obtained for all fourth-grade students aged 9–10 years enrolled in nine public elementary schools in Adachi City in 2018. Children aged 9–10 years were selected because it has been suggested that by the age of 8–10 years, children can reliably report their food intake [29]. Nine schools were selected by the local government based on their representativeness for social and geographic environments [30]. The survey consisted of two phases: the first was a breakfast survey for children, and the second was a health survey for children and their caregivers. For the breakfast survey, the questionnaires were distributed at school and children reported the foods eaten for breakfast for 7 consecutive days between September and November 2018. For the health survey, the questionnaires were distributed at school and taken home for their caregivers to complete and return in October 2018. Among the 455 child–caregiver pairs, 328 pairs provided informed consent and completed both the breakfast and health questionnaires. A total of 281 pairs were included in the present analysis, after excluding those with missing data for breakfast (*n* = 34) or mental health status (*n* = 13). We included children whose breakfast data included all 7 days because our objective was to examine what food types children should be eating for breakfast among those who ate breakfast daily. The sample comprised 131 male and 150 female children. The A-CHILD protocol and use of the data for the present study were approved by the Ethics Committee at Tokyo Medical and Dental University (Approval No. M2016-284).

### 2.2. Breakfast

Breakfast intake was assessed using a self-reported questionnaire in which the children were asked to fill in what food types they ate for breakfast over a 7-day period in a specified week. A registered dietitian coded each day’s breakfast menus as 0 or 1 for the inclusion or exclusion of six food categories according to the Japanese Food Guide Spinning Top, developed by the Ministry of Health, Labour and Welfare and the Ministry of Agriculture, Forestry and Fisheries of Japan to help people implement the Dietary Guidelines for Japanese [31]. The six food categories were: grain dishes; vegetable dishes; meat, fish, egg, and soybean dishes (main dish); milk and milk products; fruits; and confectionaries. Breads were classified according to whether they contained sweet fillings, with sweet breads containing cream, chocolate, or bean paste, or sugared breads being classified as confectionaries in accordance with the Standard Tables of Food Composition in Japan [32]. For example, if the breakfast menu was “toast, broccoli, and milk”, the categories of grain dishes, vegetable dishes, and milk and milk products were coded as 1, and the others were coded as 0. Similarly, if the breakfast menu was “chocolate bread and banana”, the categories of fruits and confectionaries were coded as 1, and the others were coded as 0. We did not account for serving sizes because the breakfast menus did not provide information on volumes. For the analysis, the average weekly frequency for each food category (total daily intakes of each category divided by seven) was used. For a detailed analysis of grain dishes, we calculated the mean frequencies for the intake of rice, bread, noodles (udon, soba, ramen, pasta), and cereals, which are typical staple foods in Japan. In addition, the children were categorized into three groups according to the frequency of rice or bread intake, as the two major staple foods in Japan. High rice eaters were defined as those who ate rice four or more times per week, high bread eaters were defined as those who ate bread four or more times per week, and all others were defined as low rice and bread eaters.

### 2.3. Mental Health Status

We evaluated the children’s difficult behaviors and prosocial behaviors as children’s mental health. The assessments of difficult behaviors and prosocial behaviors were conducted using the Japanese version of the Strength and Difficulties Questionnaire (SDQ) [33]. The SDQ is composed of 25 items and has five subscales: emotional symptoms, conduct problems, hyperactivity/inattention, peer relationship problems, and prosocial behaviors. Caregivers rated the child behaviors using a scale of 0 (not true)–2 (certainly true). The total score of four subscales (emotional symptoms, conduct problems, hyperactivity/inattention, and peer relationship problems) was calculated as the total difficult behavior score. These scores were converted into a scale of 0–100 to aid the interpretation of coefficients from the statistical analyses, as described previously [34,35]. A higher total difficult behavior score meant a higher level of difficult behaviors. A higher prosocial behavior score meant a higher level of prosocial behaviors. In our study sample, the Cronbach’s alpha values for the total difficult behavior score and prosocial behavior score were 0.82 and 0.73, respectively.

### 2.4. Covariates

Children’s statuses were assessed by a self-reported questionnaire. We considered that physical activity, sleep habits, and household socioeconomic status can be associated with both the quality of breakfast and mental health through their epidemiological relevance. Physical activity was assessed by the frequency of undertaking physical activity outside of school for ≥30 min and categorized into three groups (never/rarely, 1–4, or ≤5 times/week). Bedtime was categorized into three groups based on the children’s weekday habits (before 9:00 p.m., 9:00–9.59 p.m., and after 10:00 p.m.). Household and caregiver statuses were assessed by caregiver-reported questionnaires [36]. Household status included the household income and cohabitation status. The caregiver’s status included the mother’s age, mother’s educational attainment, mother’s employment status, and respondent’s mental health. The respondent’s mental health was assessed by the Kessler 6 scale (Japanese version) [37]. The cut-off point was a score of 5 and higher scores indicated frequent problems of psychological distress [38]. The mother’s educational attainment was categorized into three groups (low: junior high school, high school, or dropped out from high school; middle: professional school, college, or dropped out from college; or high: college or higher). The mother’s employment status was categorized into five groups (full-time, part-time, self-employed, side work, or unemployed).

### 2.5. Statistical Analysis

First, we performed a multivariate linear regression model to examine the associations between food categories for breakfast and mental health assessed as difficult behaviors and prosocial behaviors. Two models were constructed: Model 1 included individual food categories and was adjusted for potential confounders (child’s sex, physical activity, and bedtime; household income and cohabitation status; caregiver’s K6; and mother’s age, education, and employment); and Model 2 included further adjustment for all types of food categories (grain dishes, vegetable dishes, fish and meat dishes, milk and milk products, fruits, and confectionaries) to examine the relationships independently of other food categories. Second, to conduct a detailed analysis of the grain dishes, a multivariate linear regression model was used to examine the associations between the types of grain dishes and mental health. For sensitivity analyses, the frequencies of rice and bread consumption were calculated separately and their associations with mental health were analyzed. The two models described above were used. A word cloud for the children’s breakfast menus was created on the User Local website [39]. All analyses were conducted using Stata version 15 (StataCorp, College Station, TX, USA).

## 3. Results

The characteristics of the participants are presented in Table 1. Half of the participants were girls, a quarter exercised at least five times a week, more than one-third went to bed after 10:00 p.m., and approximately 90% lived with both their father and mother. The mothers were most often approximately 40 years of age, and nearly 80% were employed.

The most frequently consumed food type at breakfast was grain dishes, at six times a week (Table 2). The second most consumed food types were vegetable dishes and meat, fish, egg, and soybean dishes, with consumption once every two days. Milk products were consumed twice a week, while fruits and confectionaries were consumed once a week. In the analysis by types of grain dishes, rice and bread were consumed similarly, at three times per week. The consumption rates of noodles and cereals were low, at approximately one-tenth the rates of rice and bread. When the children were grouped by types of grain dishes, 36% ate rice at least four times per week (high rice eaters) and 39% ate bread at least four times per week (high bread eaters).

The word cloud for the breakfast menus reported by the children is shown in Appendix A. The breakfast menus were a mix of Japanese and Western dishes, with rice, bread, yogurt, salad, miso soup, milk, sausage, and eggs being commonly consumed items. The main grain dishes eaten for breakfast were rice and bread. Vegetable dishes often eaten for breakfast were tomatoes, cucumbers, broccoli, and lettuce. Among the foods categorized as meat, fish, egg, and soybean dishes, the consumption of processed foods such as sausages and ham, eggs, and natto (fermented soybeans) tended to be high. Yogurt and milk were the most frequently consumed milk products. The main fruits eaten at breakfast were bananas, oranges, and apples. Most of the foods classified as confectionaries were sweet breads or wheat-based pastries such as chocolate bread, red bean buns, donuts, and pancakes.

The associations between food categories for breakfast and mental health are shown in Table 3. The multiple linear regression analysis revealed that the more frequent consumption of grain dishes or fruits was associated with fewer problem behaviors (grain dishes: coefficient = −8.74, 95% CI: −17.0 to −0.51; fruits: coefficient = −7.63, 95% CI: −13.7 to −1.56) after the adjustment for potential confounders (Model 1). After the adjustment for all other food categories, the association for fruits was weaker (coefficient = −6.39, 95% CI: −13.1 to 0.28), but the association for grain dishes remained significant (coefficient = −12.2, 95% CI: −22.8 to −1.58) (Model 2). None of the food categories showed significant associations with prosocial behaviors.

The results of the regression analyses for mental health according to the types of grain dishes are shown in Table 4. Children who consumed rice or bread more frequently had fewer problem behaviors than those who consumed these items less frequently (high rice eaters: coefficient = −6.51, 95% CI: −10.7 to −2.27; high bread eaters: coefficient = −5.20, 95% CI: −9.25 to −1.14) (Model 2). Significant associations remained after the additional adjustment for all food categories except grain dishes. There were no significant differences between rice and bread dishes. Similar results were obtained when examining the associations between the frequencies of rice and bread intake and mental health (Appendix A).

## 4. Discussion

To the best of our knowledge, this is the first study to examine the associations between food categories eaten for breakfast and mental health in Japanese children. Grain dishes were the main breakfast category for the children and these were consumed almost daily, with most of the children consuming rice or bread as cereals. Among the food categories at breakfast, grain dishes and fruits were associated with fewer problem behaviors. Rice and bread were both associated with a lower risk of problem behaviors.

We found that rice and bread were the two major grain dishes eaten at breakfast by the children. This pattern is consistent with a study on the breakfast patterns in Japanese adults, which found four types of patterns, including two staple meal patterns, as follows: rice-based (rice/vegetable/pulse/seasoning), bread-based (bread/dairy/fruit/sugar), meat/egg/fat, and tea/coffee [23]. Another study on Japanese university students found that the mean frequency of rice or bread intake at breakfast was three times a week, similar to the present findings [40]. The consumption of grain dishes other than bread and rice, i.e., noodles and cereals, was low, at approximately one-tenth the rates of rice and bread. Instead, children often consumed sweet breads categorized as confectionaries. Therefore, breakfast patterns other than bread or rice may be sweet bread patterns for children in Japan.

The present finding that the more frequent consumption of grain dishes, whether rice or bread, at breakfast was associated with fewer problem behaviors in children is plausible, because grain dishes are rich in carbohydrates that can be converted into glucose, which is primary fuel for the brain. The brain is sensitive to fluctuations in glucose supply, and the maintenance of adequate blood glucose concentrations between meals is thought to be beneficial for optimal cognition [11]. A study of children aged 9–11 years found improvements in memory and task performances after the consumption of glucose-containing drinks [41]. Therefore, the consumption of grain dishes at breakfast may have been effective for the supply of glucose after the overnight fast. Meanwhile, biological findings have indicated that carbohydrate intake increased the brain uptake of tryptophan from the plasma, leading to the synthesis of serotonin in the brain [42]. Serotonin, a neurotransmitter, plays an important role in mood alleviation [43]. Therefore, eating carbohydrate-rich grain dishes for breakfast may increase tryptophan bioavailability in the central nervous system, leading to the alleviation of mood.

In terms of glucose, many foods classified as confectionaries, such as chocolate bread, cream buns, donuts, and sugar toast, were also glucose-rich dishes. However, the consumption of confectionaries was not associated with problem behaviors in this study. One explanation may be the difference in glycemic load (GI). Compared with grain dishes high in polysaccharides, confectionaries are high in monosaccharides and have a higher GI. Lower GI values can minimize blood glucose fluctuations, and lower GI breakfasts were reported to be associated with better attention and less frustration in children compared with high-GI breakfasts [44]. A recent review found that sugar intake induced multiple physiological responses, including systemic inflammation, dopamine signaling disorders, oxidative stress, and insulin resistance, all of which are associated with depression [45]. Given these findings, it is possible that mental health is better served by the intake of non-sweet carbohydrates over sweet carbohydrates at breakfast.

The frequent consumption of fruits at breakfast was also associated with fewer problem behaviors in children. This finding is in line with the results of several observational studies on children, although breakfast was not evaluated [45,46]. A study in Australia showed that children with problem behaviors consumed fewer servings of fruits than children without problem behaviors [45]. Another study conducted on adolescents found that significantly fewer problem behaviors were observed with the increased intake of fresh fruits [46]. In the indicated studies, vegetable dishes and fruits were both associated with better mental health [45,46], but these associations were not observed in the present study. In our study, the association between fruits and mental health was no longer significant after the adjustment for other food categories including grain dishes. These findings suggest that the fructose in fruits may have contributed to problem behaviors or that other foods consumed with fruits may have had an effect (confounding).

Our study has some limitations. First, the validity and reliability of the breakfast measurement method in children aged 9–10 years have not been confirmed. Instead of estimating detailed foods and nutrients from the foods described by the children, we devised a way to achieve the measurements with high accuracy by only assessing the presence or absence of food category intakes. Second, we were unable to assess the quantities of each food category eaten for breakfast. Meanwhile, for foods associated with multiple food categories, we were often unable to identify the ingredients present in the foods. For example, some children did not report the details of the ingredients in their soups and stews. Therefore, the intake of vegetable dishes and meat, fish, egg, and soybean dishes, which are often included as ingredients, were often not counted as intake, possibly leading to underestimations in the results. In the future, it is necessary to devise methods that encourage children to describe the foods in detail and to obtain evaluations from caregivers to examine the validity of the method. Third, we were unable to evaluate dietary data other than breakfast intakes. However, the children were offered the same nutritious lunch at school during the day. Finally, the generalizability of the present results is limited because meals vary from culture to culture. The children in the present study ate rice or bread dishes as their main breakfast foods, had vegetable dishes and meat/fish/egg/soybean dishes every other day, and consumed fruits less frequently. Studies in other areas are needed. Despite these limitations, the fact that we were able to obtain breakfast data for children over a 7-day period provided very valuable evidence.

## 5. Conclusions

Elementary school children in Japan consumed more rice and bread dishes for breakfast compared with other food categories. Vegetable dishes and meat/fish/egg/soybean dishes were consumed every other day, and fruits and confectionaries were consumed about once a week. Children who consumed grain dishes or fruits had fewer problem behaviors than children who did not. Grain dishes, whether rice or bread, were associated with reduced problem behaviors, while confectionaries, which mainly consisted of sweet breads or pastries, were not associated with problem behaviors. These results should aid in the development of dietary recommendations for breakfast among Japanese children. Although further examinations are warranted in longitudinal and intervention studies, the intake of non-sweet grain dishes or fruits at breakfast may be effective for preventing behavioral problems in children. This study could add evidence that it is not only whether children eat breakfast but also the type of food they eat for breakfast matters. The public health impact of this study is to make caregivers aware of the benefits of having their children eat grain dishes or fruits for breakfast. Neither grain foods nor fruits require complex preparation and are therefore easy to prepare at home. Few schools offer breakfast in Japan, and the food contents of breakfasts served in the “school breakfast program” in Japan were shown to vary from school to school [47]. In the future, serving grain dishes and fruits may contribute to improvements in children’s mental health.

## Figures and Tables

**Table 1 nutrients-15-01091-t001:** Characteristics of the Japanese school children (*n* = 281).

			*n*	%
Child’s status				
	Sex			
		Male	131	46.6
		Female	150	53.4
	Physical activity			
		Never/rarely	27	9.6
		1–4 times/weeks	182	64.8
		≤5 times/weeks	72	25.6
	Bedtime			
		Before 9:00 p.m.	20	7.1
		9:00–9:59 p.m.	155	55.2
		After 10:00 p.m.	105	37.4
		Missing	1	0.4
	Type of grain dishes			
		High rice eater	101	35.9
		High bread eater	109	38.8
		Low rice and bread eater	71	25.3
Household status				
	Household income (million JPY)			
		<3.00	23	8.2
		3.00–5.99	88	31.3
		6.00–9.99	94	33.5
		≥10.0	39	13.9
		Missing	37	13.2
	Cohabitation status			
		Parents	249	88.6
		Mother	25	8.9
		Father	3	1.1
		No parent	4	1.4
Caregiver’s status				
	Mother’s age (years)			
		<35	20	7.1
		35–44	186	66.2
		≥45	72	25.6
		Missing	3	1.1
	Mother’s education			
		Low	49	17.4
		Middle	58	20.6
		High	43	15.3
		Other/missing	131	46.6
	Mother’s employment status			
		Full-time	67	23.8
		Part-time	128	45.6
		Self-employed	19	6.8
		Side work	4	1.4
		Not employed	59	21
		Other/missing	4	1.4
	Respondent’s K6			
		<5	203	72.2
		≥5	77	27.4
		Missing	1	0.4
Child’s behavior problems (SDQ score) ^a^			Mean	SD
	Total difficulties score		22.7	14.1
	Emotional symptoms		18.8	18.9
	Conduct problems		22.5	18.4
	Hyperactive/inattention		31.5	23.1
	Peer relationship problems		18.2	17.7
	Prosocial behavior		68.5	21.3

K6, Kessler 6; SD, standard deviation; SDQ, Strength and Difficulties Questionnaire. ^a^ All variables ranged from 0 to 100.

**Table 2 nutrients-15-01091-t002:** Breakfast intakes by the Japanese school children (*n* = 281).

		Mean	SD
Child’s food category intake at breakfast (*n*/week)			
	Grain dishes	6.02	1.39
	Vegetable dishes	4.17	2.40
	Meat, fish, egg, and soybean dishes	3.27	2.13
	Milk and milk products	1.90	2.35
	Fruits	1.16	1.90
	Confectionaries	0.95	1.46
Child’s intake of food type of grain dishes at breakfast (*n*/week)			
	Rice	2.77	2.06
	Bread	2.96	2.03
	Noodle	0.28	0.70
	Cereal	0.22	0.79

**Table 3 nutrients-15-01091-t003:** Results of regression analyses on children’s mental health according to the frequencies of food categories for breakfast among Japanese school children (*n* = 281).

Food Category		Behavior Problems (SDQ Score)	Prosocial Behavior (SDQ Score)
		Coefficient (95% CI)	Coefficient (95% CI)
Model 1			
	Grain dishes	−8.74 (−17.0 to −0.51)	−2.45 (−15.5 to 10.59)
	Vegetable dishes	−0.47 (−5.49 to 4.54)	−0.61 (−8.5 to 7.27)
	Meat, fish, egg, and soybean dishes	−1.98 (−7.48 to 3.53)	−4.67 (−13.3 to 3.97)
	Milk and milk products	−4.56 (−9.5 to 0.44)	−0.07 (−8.0 to 7.82)
	Fruits	−7.63 (−13.7 to −1.56)	5.70 (−3.9 to 15.32)
	Confectionaries	0.58 (−7.31 to 8.47)	−2.43 (−14.8 to 9.97)
Model 2			
	Grain dishes	−12.2 (−22.8 to −1.6)	−5.61 (−22.6 to 11.38)
	Vegetable dishes	0.10 (−5.33 to 5.52)	0.77 (−7.9 to 9.45)
	Meat, fish, egg, and soybean dishes	−0.11 (−6.24 to 6.03)	−5.07 (−14.9 to 4.74)
	Milk and milk products	−2.51 (−7.96 to 2.94)	−1.23 (−10.0 to 7.49)
	Fruits	−6.39 (−13.1 to 0.28)	7.54 (−3.1 to 18.20)
	Confectionaries	−4.01 (−14.3 to 6.23)	−8.46 (−24.9 to 7.93)

CI, confidence interval; SD, standard deviation; SDQ, Strength and Difficulties Questionnaire. All variables ranged from 0 to 100. Model 1: Individual food categories with adjustment for child’s sex, physical activity, and bedtime, household income and cohabitation status, caregiver’s K6, and mother’s age, education, and employment. Model 2: Model 1 with adjustment for all types of food categories (grain dishes, vegetable dishes, fish and meat dishes, milk and milk products, fruits, and confectionaries).

**Table 4 nutrients-15-01091-t004:** Results of regression analyses on children’s mental health according to types of grain dishes among Japanese school children (*n* = 281).

Food Category		Behavior Problems (SDQ Score)	Prosocial Behavior (SDQ Score)
		Coefficient (95% CI)	Coefficient (95% CI)
Model 1			
	High rice eater	−6.51 (−10.7 to −2.27)	1.48 (−5.3 to 8.26)
	High bread eater	−5.20 (−9.25 to −1.14)	2.80 (−3.7 to 9.28)
	Low rice and bread eater	Referent	Referent
Model 2			
	High rice eater	−7.74 (−12.3 to −3.19)	1.63 (−5.7 to 9.01)
	High bread eater	−5.27 (−9.62 to −0.92)	2.05 (−5.0 to 9.10)
	Low rice and bread eater	Referent	Referent

CI, confidence interval; SD, standard deviation; SDQ, Strength and Difficulties Questionnaire. All variables ranged from 0 to 100. Model 1: Adjusted for child’s sex, physical activity, and bedtime, household income, and cohabitation status, caregiver’s K6, and mother’s age and employment. Model 2: Model 1 with further adjustment for all types of food categories except grain dishes (vegetable dishes, fish and meat dishes, milk, and milk products, fruits, and confectionaries).

## Data Availability

The datasets generated and analyzed during the current study are not publicly available due to ethical or legal restrictions but are available from the corresponding author on reasonable request.

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
