# Peer review of "Food Categories for Breakfast and Mental Health among Children in Japan: Results from the A-CHILD Study"

_nutrients, 2023, doi:10.3390/nu15051091_

Round 1

Reviewer 1 Report

The subject of the study is very interesting and the result is useful.

The authors examined the breakfast habits and mental status of 281 young school-aged children. More precisely, they investigated the effect of the nutritional content on the mental status of children.

The result is very interesting.

We think that nutrients with a sweet taste and content are more important for children, if they eat them, they also calm down mentally.

The study does not prove this. The study shows that it is better for the mental status of school children if they eat bread and rice.

The research was carried out using a carefully developed method, the children prepared the prescribed foods for 7 days based on the prescribed category.

The manuscript is well edited, its content and language are understandable.

Editing of the abstract is recommended:

- introduction

- Aim/Purpose

- Methods

- Discussion

- Conclusion parts.

The manuscript contains four tables and illustrates the results of the research with excellent statistical data processing.

The four authors are employees of one institute but two departments.

The manuscript refers to 46 references, of which 40 have DOI identifiers, one has htts internet access.

The study examines a rare area, influencing the mental life of young people through nutrition is a very useful result.

The implementation of the result can have a long-term effect.

Congratulations to the authors.

Well done

After revising the abstract, the manuscript is recommended for acceptance and publication.

Author Response

1 February 20, 2023 Response to Reviewers Re: nutrients-2240495, Food categories for breakfast and mental health among children in Japan: Results from the A-CHILD study We thank the reviewer for their helpful comments. We have revised the manuscript by addressing the comment below. 1. In the introduction, it is worth adding that the so-called functional food. This food is easy to prepare and healthy at the same time. This type of food includes, for example, extruded products (e.g. breakfast cereals, co-extruded cereals, functional breadcrumbs, etc.). It is this ease of preparation that sometimes determines the preparation of a quick and tasty breakfast. Check out the following article: "Characterization of corn extrudates with the addition of brewers' spent grain as a raw material for the production of functional batters". Another example article is: Functional foods: Product development, technological trends, efficacy testing, and safety. I think this is very important in the context of your paper. Reply: Thank you for this comment. Since our study does not examine functional foods, we are wondering if it would not be appropriate to add about functional foods in the introduction of the paper. Would it be possible for you to advise us in more detail as to why functional foods should be added? 2. I think you should better justify the idea behind your article. Answer the question - why are your studies so important? This will highlight the innovation of your research. The science goal seems good. Reply: Thank you for this helpful comment. We have added the following texts to the Introduction section: 2 “There is consensus on the importance of eating breakfast; however, there is limited evidence that what children eat for breakfast contributes to their mental health. The goal of this study is to contribute to the improvement of children’s mental health by providing evidence about what to eat for breakfast to prevent problem behaviors in children.” 3. Write why you chose a group of children aged 9-10. In fact, you can justify it anywhere and in any way - maybe somewhere in the methodology? Reply: Thank you for this important comment. We have added the following texts and reference to the Method section: “Children aged 9-10 years were selected because it has been suggested that by the age of 8-10 years, children can reliably report their food intake. [29]” 4. Statistical analysis: Write what program you used to perform the analysis, device name: manufacturer, city, country. Reply: Thank you for this important comment. We have added the following information to the Method section: “All analyses were conducted using Stata version 15 (StataCorp, TX, USA).” 5. Conclusion: I think you should expand your conclusions more. Add more prospective conclusions. Answer the question: What does your research contribute to improving children's health? Will your research lead to a better selection of breakfast items, etc. Reply: Thank you for this important comment. We have added the following texts to the Conclusion: 3 “Elementary school children in Japan consumed more rice and bread dishes for breakfast compared with other food categories. Vegetable dishes and meat/fish/egg/soybean dishes were consumed every other day, and fruits and confectionaries were consumed about once a week. Children who consumed grain dishes or fruits had fewer problem behaviors than children who did not. Grain dishes, whether rice or bread, were associated with reduced problem behaviors, while confectionaries, which mainly consisted of sweet breads or pastries, were not associated with problem behaviors. These results should aid in the development of dietary recommendations for breakfast among Japanese children. Although further examinations are warranted in longitudinal and intervention studies, intake of non-sweet grain dishes or fruits at breakfast may be effective for preventing behavioral problems in children. This study could add evidence that not only whether children eat breakfast but also the type of food they eat for breakfast matters. The public health impact of this study is to make caregivers aware of the benefits of having their children eat grain dishes or fruits for breakfast. Neither grain foods nor fruits require complex preparation and are therefore easy to prepare at home. Few schools offer breakfast in Japan, and the food contents of breakfasts served in the “school breakfast program” in Japan were shown to vary from school to school.[46] In the future, serving grain dishes and fruits may contribute to improvements in children’s mental health.” Again, thank you for these very kind and helpful comment. We hope that the paper is now suitable for publication in Nutrients. Sincerely, Yukako Tani Aya Isumi Satomi Doi Takeo Fujiwara

Reviewer 2 Report

Dear Authors,

Overall, this is a good and interesting paper. In my opinion, you have also described a very important problem, which is the mental health of children, who will become adults in the future. The correlation of these issues with the consumption of breakfast (breakfast products) is also interesting. To improve the quality of this article, please review the comments below.

In the introduction, it is worth adding that the so-called functional food. This food is easy to prepare and healthy at the same time. This type of food includes, for example, extruded products (e.g. breakfast cereals, co-extruded cereals, functional breadcrumbs, etc.). It is this ease of preparation that sometimes determines the preparation of a quick and tasty breakfast. Check out the following article: "Characterization of corn extrudates with the addition of brewers' spent grain as a raw material for the production of functional batters". Another example article is: Functional foods: Product development, technological trends, efficacy testing, and safety. I think this is very important in the context of your paper.

I think you should better justify the idea behind your article. Answer the question - why are your studies so important? This will highlight the innovation of your research. The science goal seems good.

Write why you chose a group of children aged 9-10. In fact, you can justify it anywhere and in any way - maybe somewhere in the methodology?

Statistical analysis: Write what program you used to perform the analysis, device name: manufacturer, city, country.

Conclusion: I think you should expand your conclusions more. Add more prospective conclusions. Answer the question: What does your research contribute to improving children's health? Will your research lead to a better selection of breakfast items, etc.

Author Response

(The authors gave the same response as above.)
